# Rhizosphere Fungal Dynamics in Sugarcane during Different Growth Stages

**DOI:** 10.3390/ijms24065701

**Published:** 2023-03-16

**Authors:** Qiang Liu, Ziqin Pang, Yueming Liu, Nyumah Fallah, Chaohua Hu, Wenxiong Lin, Zhaonian Yuan

**Affiliations:** 1National Engineering Research Center for Sugarcane, Fujian Agriculture and Forestry University, Fuzhou 350002, China; 2College of Agricultural, Fujian Agriculture and Forestry University, Fuzhou 350002, China; 3Fujian Provincial Key Laboratory of Agro-Ecological Processing and Safety Monitoring, College of Life Sciences, Fujian Agriculture and Forestry University, Fuzhou 350002, China; 4Key Laboratory of Crop Ecology and Molecular Physiology, Fujian Agriculture and Forestry University, Fuzhou 350002, China; 5Province and Ministry Co-Sponsored Collaborative Innovation Center of Sugar Industry, Nanning 530000, China

**Keywords:** rhizosphere fungi, temporal variation, soil properties, structural equation modeling, fungi community assembly

## Abstract

Understanding the normal variation of the sugarcane rhizosphere fungal community throughout its life cycle is essential for the development of agricultural practices for fungal and ecological health associated with the microbiota. Therefore, we performed high-throughput sequencing of 18S rDNA of soil samples using the Illumina sequencing platform for correlation analysis of rhizosphere fungal community time series, covering information from 84 samples in four growth periods. The results revealed that the sugarcane rhizosphere fungi possessed the maximum fungal richness in Tillering. Rhizosphere fungi were closely associated with sugarcane growth, including Ascomycota, Basidiomycota, and Chytridiomycota, which showed high abundance in a stage-specific manner. Through the Manhattan plots, 10 fungal genera showed a decreasing trend throughout the sugarcane growth, and two fungal genera were significantly enriched at three stages of sugarcane growth (*p* < 0.05) including *Pseudallescheria* (Microascales, Microascaceae) and *Nectriaceae* (Hypocreales, Nectriaceae). In addition, soil pH, soil temperature, total nitrogen, and total potassium were critical drivers of fungal community structure at different stages of sugarcane growth. We also found that sugarcane disease status showed a significant and strong negative effect on selected soil properties by using structural equation modeling (SEM), suggesting that poor soil increases the likelihood of sugarcane disease. In addition, the assembly of sugarcane rhizosphere fungal community structure was mainly influenced by stochastic factors, but after the sugarcane root system became stable (Maturity), the stochastic contribution rate decreased to the lowest value. Our work provides a more extensive and solid basis for the biological control of sugarcane potential fungal diseases.

## 1. Introduction

The microbial community in the rhizosphere environment is critical for the health of terrestrial plants and sustainable soil development [1]. As more attention is paid to the relationship between plants and soil microbiomes, comprising the interactions between microbial and plant root systems and genetic components [2,3,4], it is apparent that more exploration is needed of what constitutes normal temporal variations in rhizosphere microbial community structures and compositions. In particular, research related to rhizosphere fungal communities that may produce soil-borne diseases of plants is essential. Variations in the rhizosphere fungal group and between different plants and life-cycle stages have a lot of space for exploration. As we all know, after germinating in the soil, plant roots begin to recruit microbial communities closely related to their growth through the accumulation of root secretions, rhizosphere deposition, and nutrient uptake. Thus, the growth activities of plants during different periods cause changes in rhizosphere fungal communities and functions. For instance, model plants (*Arabidopsis thaliana* and *Medicago truncatula*) can retain certain fungal populations in the soil through certain chemicals secreted by their root systems [5]. In addition, advances in sequencing technology provide a foundation for research on the dynamic changes of microbes in the rhizosphere of plants in space and time [6]. Similarly, these types of studies are needed to understand the immigration and emigration patterns of microorganisms between different parts of the plant, between different points of time and between different plants and soil environments [7,8,9], especially the related research on time series. For example, Zhang et al. (2018) identified biomarker taxa and established a model to correlate root microbiota with rice resident time in the field using a machine learning approach [10], and Li et al. (2020) analyzed the variation of the fungal community structure in the rhizosphere of *Gardenia*, which provided a new theoretical framework for further study on the rhizosphere mechanism [11].

Furthermore, as an important economic crop of food and biofuel, sugarcane is constantly expanding its planting area under the condition of global warming [12,13]. The study of sugarcane rhizosphere fungi temporal variation has great significance in better understanding the interaction between sugarcane and rhizosphere microbiota [14]. For example, sugarcane growth is facing a huge threat due to Yellow Canopy Syndrome (YCS) in Australia. YCS is a kind of largely undiagnosed plant disease that is impacting sugarcane growth across Queensland, Australia, causing huge yield losses [15]. Studying the temporal variation of sugarcane rhizosphere microorganisms provides a new way of thinking about the solution of YCS. Additionally, given the growing importance of sugarcane [16,17], identifying ways to promote the healthy growth of sugarcane and sustainably increase productivity is critical, and hence, the potential to harness microorganisms from the sugarcane rhizosphere has recently gained more attention [18,19,20]. For instance, rhizosphere microorganisms are involved in the solubilization of phosphorus and potassium containing minerals [21]. Sugarcane smut disease, caused by a fungus called *Sporisorium scitamineum* (Basidiomycota, Ustilaginales), is a limiting factor to cane production and a potential threat to the sugar industry. Juma and Musyimi’s research showed that the selected isolates from sugarcane rhizosphere microflora had an evidently antagonistic activity against the *Sporisorium scitamineum*, and is recommended as a potential biocontrol agent for this pathogen [22]. In a related study, She et al. (2021) found maize rhizosphere microorganisms are critical for facilitating microbiome bioremediation for soil affected by neutral-alkaline mining [23]. Moreover, understanding how the sugarcane rhizosphere fungal population change with stages would provide valuable data and could in the long-term result in the identification of potential management strategies for healthy cultivation of sugarcane, including an improved breeding process taking the microbe into account or better targeted biological prevention and treatments [24]. Here, we used sugarcane as an experimental plant to fill critical knowledge gaps that will accelerate our exploration process to successfully harness rhizosphere fungal groups to sustainably enhance sugarcane productivity. This also has reference significance for other crops in nature. Our study had three main objectives: (i) to determine the drivers of the sugarcane fungal community under field conditions; (ii) to understand the seasonal trends of fungi in the rhizosphere of sugarcane; and (iii) to explore the assembly and interaction relationships of rhizosphere fungal communities during sugarcane development. To achieve this, we examined fungal community assemblages of sugarcane rhizosphere soil through Illumina amplicon sequencing of the samples collected from young to mature sugarcane (Seeding, Tillering, Elongation, and Maturity). Our results provide more foundational insight into the process of sugarcane root microbiota exploration.

## 2. Results

### 2.1. Sugarcane Rhizosphere Nutrients Produce Differential Changes at Different Stages

The results showed that the height and stem diameter of sugarcane plants increased continuously over time, with the rate of diseased plants reaching their highest (19.8%) in Elongation (Table 1). We also investigated the temporal variation of soil biochemical properties collected in Seeding, Tillering, Elongation, and Maturity during the different growth stages of sugarcane. Our analysis showed that soil biochemical properties varied with time. Soil properties such as soil pH and available nitrogen (AN) decreased considerably (*p* < 0.05), with Elongation and Maturity recording the lowest soil pH, while Seeding had recorded the highest, followed by Tillering. In addition, the soil organic matter (OM) and total phosphorus (TP) were significantly higher in Elongation compared to Seeding, Tillering, and Maturity (*p* < 0.05). We also observed that Seeding recorded a significant amount of soil total nitrogen (TN) relative to the other stages. Compared to Elongation, a considerable amount of soil available phosphorus (AP) was recorded in Seeding and Tillering, and the available phosphorus nutrient decreased by 20.07% and 25.37%, respectively. In comparison to Tillering and Maturity, soil total potassium (TK) revealed significant improvement (*p* < 0.05) in Seeding, followed by Elongation. However, soil available potassium (AK) also showed significant differences among the different time points, with the lowest AK values in Maturity (*p* < 0.05), which is significantly reduced by 24.68% compared with the highest value. Soil temperature peaked significantly in Tillering and Elongation than in Seeding and Maturity (Table 2).

### 2.2. Rhizosphere Fungal Diversity Differs between Different Stages and Correlates with Soil Properties

The total number of sequences in the raw sequencing data is 4,592,374 with an average length of 401.21. The shortest sequence length is 221 and the longest sequence length is 453. For details, see Annexes-1. Fungal diversity (Shannon and Simpson), richness (ACE), and Sobs (number of observed species) were assessed during the different time points. It was observed that fungal diversity increased substantially (*p* < 0.05) in Tillering compared with the other stages (Table 3 and Appendix A). Moreover, compared to Maturity, fungal richness and Sobs were significantly enriched (*p* < 0.05) in Tillering, followed by Seeding and Elongation (Table 3, Appendix A). In addition, the dilution curve showed a sufficient amount of sequencing data (Appendix A).

We carried out Principal Coordinates Analysis (PCoA) to explore and visualise similarities or dissimilarities in fungal community composition in all the samples collected during the different growth stages of sugarcane. Based on the growth changes of FN 41 sugarcane throughout its life cycle, simulation map with distinctive features for different stages were constructed (Figure 1A). The analysis revealed that fungal community composition was clustered together in the different samples, and the soil samples shifted with different stages in the first axis (Figure 1B).

The relationship between soil environmental variables and fungal alpha diversity was investigated using Pearson’s correlation coefficients (Figure 2A). The analysis demonstrated that soil temperature (Tem) exhibited a strong and positive association with fungal Sobs and ACE, Shannon and PC1. However, soil Tem was negatively associated with PC2. Furthermore, soil pH revealed a significant positive correlation with PC1, Sobs, and ACE. Soil TN and AN were significantly and positively associated with PC1. Soil TN also exhibited a positive relationship with PC2, while TK had a strong positive association with PC1 and PC2. We also noticed that soil TP demonstrated a positive correlation with PC2. On the other hand, soil TN and TK were negatively related to Shannon. Meanwhile, regression analysis was conducted to further confirm the association among soil environmental variables, fungal ACE (richness), Shannon (diversity), PC1 and PC2 during the different time points (Appendix A). The analysis demonstrated that soil pH and AN showed a positive relationship with fungal richness. Moreover, soil Tem exhibited a positive relationship with fungal richness, particularly in Tillering and Elongation (Appendix A). Soil pH and C/N showed a positive correlation with fungal richness in Tillering, whereas AN showed a strong positive correlation with fungal richness in Seeding. In Tillering and Elongation, soil Tem revealed a strong positive relationship with fungal richness (Appendix A). Furthermore, soil pH, TN, AP, AN, and AK had positive relationships with PC1 in Seeding, while in Tillering and Elongation, soil Tem was positively related to PC1 (Appendix A), whereas in Seeding, TN and TK had a positive association with PC2, while in Elongation and Maturity soil TP and AK demonstrated a positive correlation with PC2, respectively (Appendix A).

### 2.3. Variability in Fungal Community Composition at Different Growth Stages

Fungal relative abundance during the stages was assessed at the phylum level. The analysis showed that Ascomycota accounted for more than 50% of the absolute dominance in the entire sample present in the rhizosphere soil of sugarcane, while Basidiomycota, Chytridiomycota, and Glomeromycota were present in smaller dominant groups in the hot zone of sugarcane roots. However, Chytridiomycota and Glomeromycota were significantly higher (*p* < 0.05) in Tillering than the other stages (Appendix A). Basidiomycota increased significantly in Tillering and Seeding compared to other phases and the mean sample abundance peaked in Tillering (Appendix A). Venn diagram analysis demonstrated that 15, 23, 11, and 11 unique fungal OTUs were detected in Seeding, Tillering, Elongation, and Maturity, respectively (Figure 3B). Moreover, among the soil samples collected in the various stages, 451 OTUs were shared among the different time points. In Tillering, the highest OTUs were recorded compared with the other sampling times.

### 2.4. Fungal Community Composition, FUNGuild, and Soil Physicochemical Properties Are Interrelated

The different environmental variables were significantly correlated with the variation of selected soil fungal genera (Figure 4A). RDA demonstrated the relationship between soil properties and fungal communities, which had an eigenvalue of 0.2846 for the first axis and 0.0802 for the second axis, respectively. The vectors indicated the following: total N, total P, total K, pH, available N and soil temperature played a greater role than available K, available P and organic matter for sugarcane rhizosphere fungi genera. The soil fungus community was dominated by Basidiomycota and Ascomycota, which showed stronger associations with higher pH, available N and soil temperature, while Chytridiomycota slightly revealed the opposite trend. We then used the Source Model of Plant Microbiome (SMPM) to estimate the proportion of sugarcane rhizosphere fungal communities from “adjacent period” and “unknown” sources and took the adjacent periods as the source and library in turn (Figure 4B). The results showed that the related fungal communities mainly came from the transmission of adjacent periods, and the transfer proportion decreased gradually with the migration of time. In addition, Tremellales and Saccharomycetales, as the dominating orders in fungi, were positively associated with pH and AN. However, Sordariomycetes and Pseudallescheria showed a stronger negative association with pH and AN (Figure 4D). To further explore the environmental driving factors of the sugarcane fungus community, we conducted a Mantel-test analysis and analyzed the correlation between the microbial matrix and the soil property matrix, and then we correlated distance-corrected dissimilarities of taxonomic and functional community composition with those of environmental factors. Overall, pH, TN and TK exhibited the strongest correlations with the taxonomic composition in the sugarcane rhizosphere soil (Figure 4C), while no significant correlation was found for other soil factors (*p* < 0.05). In addition, soil temperature and pH were only weakly correlated with taxonomic and functional community composition. Besides, almost all the environmental variables exhibited no significant correlations, except for AN.

### 2.5. Fungal Communities Differ in Co-Occurrence Networks with Changes in Growth Periods

We employed Manhattan plots to examine fungal community composition OTUs differences in sugarcane rhizosphere soil during the growth stages (Figure 5 and Appendix A). The analysis revealed that Basidiomycota, Ascomycota, and Chytridiomycota were significantly more abundant in Tillering than in Seeding (Figure 5A). Additionally, Ascomycota increased considerably in Tillering relative to Seeding; however, Basidiomycota and Chytridiomycota decreased profoundly in Seeding (Figure 5B). In Elongation, Basidiomycota and Ascomycota increased significantly, while Chytridiomycota diminished considerably in Maturity (Figure 5C). Meanwhile, Venn diagram analysis was adopted to gain a deeper understanding of the genera of fungal OTUs from one time point to another. The analysis demonstrated that between Seeding and Tillering, the highest number of enriched fungal OTUs was detected, followed by Elongation to Maturity (Figure 5E). However, between Tillering and Elongation the highest amount of depleted fungal OTUs were identified, followed by Elongation to Maturity (Figure 5D). Meanwhile, the co-occurrence network showed the interactions of rhizosphere fungi in sugarcane during four critical periods, where the fungi belonging to mainly *Talaromycetes*, *Talaromyces*, *Fusarium*, *Sordariales*, and *Pseudallescheria* were dominated (Figure 5F). The role of each fungus in the network changed over time during different reproductive periods, with the degree and importance changing. Among the four critical fertility networks, Seeding had the largest average density and mean degree, followed by Maturity (Appendix A).

### 2.6. Construction of Structural Equation Models Related to Pokkah Boeng Disease of Sugarcane

We constructed an SEM to assess the direct and indirect effects of soil properties (pH, AN and TN), microbial genera (*Fusarium* and *Talaromyces*), and microbial diversity (ACE and Shannon) on the incidence of sugarcane (Figure 6 and Appendix A and Appendix A). We used a multigroup modelling method to assess which relationships exist between soil properties, microbial communities, and sugarcane disease during the growing process. The results of model-1 showed that soil pH was significantly and directly affected the abundance of *Fusarium,* sugarcane disease and fungal diversity. However, soil pH negatively regulated the disease rate of sugarcane (the lower the pH, the more disease occurs) (Figure 6A and Appendix A). Meanwhile, AN, TN and disease had similar significant regulatory relationships as pH and disease (Appendix A, Appendix A). In addition, soil temperature affected the diversity of rhizosphere fungi, and had both direct and indirect effects on the abundance of *Fusarium* and *Talaromyces.*

### 2.7. The Assembly Process of Fungal Communities Is Influenced by Changes in the Growth Period

The results showed that the mean nearest-taxon index (NTI) was greater than 0, and the mean nearest taxon distance between samples (β-NTI) values among the samples from the critical fertility period of sugarcane were mainly concentrated in the interval of −2 to 2, indicating that the changes in microbial community structure during the critical fertility period were mainly influenced by stochastic factors. The existence of a certain number of samples with β-NTI values greater than 2 among the soils in different periods indicated that the changes in the rhizosphere microbial community structure of some soil samples were mainly influenced by deterministic factors (Figure 7B). In addition, the β-NTI values of samples between stages (Seeding–Tillering, Tillering–Elongation, and Elongation–Maturity) were between (−2, 2), indicating that the changes in the rhizosphere microbial community structure of sugarcane with time were mainly influenced by stochastic factors, but there were differences in the contribution of stochastic factors with time (Appendix A).

### 2.8. Functional and Evolutionary Relationships among the Major Fungal Genera of Rhizosphere Soil

As for functional classification, the fungal communities in rhizosphere soil were classified by using the trophic mode (Figure 8). A maximum likelihood phylogenetic tree was conducted to further visualized the relationship among crucial fungal communities. Following the procedure, the top 28 genera, were classified into five guilds. They were divided into Undefined Saprotroph, Dung Saprotroph, Animal Pathogen, Plant Pathogen, and Endophyte, respectively. Based on the classification, the results revealed that 33.3% of these genera belonged to Undefined Saprotroph, 3.3 % belonged to Plant Pathogen, 6.6 % belonged to Animal Pathogen, 6% belonged to Dung Saprotroph, 3.3% belonged to Endophyte while 46.6% of these genera were unclassified. We also observed that these top 28 genera, were detected in 8 different phyla, of which five were identified as Ascomycota, Chytridiomycota, Glomeromycota, and Basidiomycota, respectively. Moreover, the plant pathogenic fungi observed in this study was *Cochliobolus,* and it is worth noting that the abundance of *Fusarium* genera was higher in Tillering and Maturity. In addition, according to the results of the FUNGuild classification difference (Appendix A), in the comparison of the Seeding and Tillering of sugarcane growth (Appendix A), there were more classifications with significant differences (*p* < 0.05). They were Dung Saprotroph-Soil Saprotroph-Wood Saprotrop, Animal Pathogen–Endophyte–Lichen Parasite–Plant, Orchid Mycorrhizal–Plant Pathogen–Wood Saprotroph, Endophyte–Plant Pathogen, Animal Pathogen–Soil Saprotroph, respectively. However, with the shift in sugarcane stages, the number of significantly different FUNGuild classifications gradually decreased, accompanied by a decrease in the significance of differences between some FUNGuild classifications. (Appendix A).

## 3. Discussion

### 3.1. Changes in Sugarcane Growth Periods Alter the Composition and Diversity of Soil Properties and Fungal Communities

Soil microbial communities associated with plants can have strong influences on plant growth as well as contribute to soil health and sustainable production [25,26,27]. Therefore, understanding the temporal progression of rhizosphere microbiota is a prerequisite for plant and soil environmental improvement [28,29]. Previous research showed that plants’ root microbiota composition varied with plant developmental stage [30,31], but these studies were carried out either with other model crops or under greenhouse conditions. Our findings provide a detailed description of the rhizosphere fungal population during the entire sugarcane growth period in the field, as well as insights into how sugarcane growth and the soil environment influence the development of the rhizosphere fungal population. We showed temporal shifts in fungal community composition during the life of sugarcane, and these results are missing in sugarcane-related research in the field (Figure 1A,B). Our findings showed that the nutrients in sugarcane rhizosphere soil exhibited significant changes during different growth periods (*p* < 0.05), which we assumed was precipitated by the difference in sugarcane requirements for different nutrients at different growth stages [32]. Moreover, although the alpha diversity of the sugarcane rhizosphere fungal community varies significantly between different growth periods, it has a gradually stable trend (Appendix A). It showed that sugarcane has the ability to regulate its own rhizosphere fungal environment to maintain a stable state under the natural growth conditions. Whether this is applicable to other crop systems requires more extensive and in-depth studies. Additionally, the species composition showed that rhizosphere fungi were mainly Ascomycota and Basidiomycota during the growth period of sugarcane (Figure 3A), which was similar to the research data of Zeng et al. (2020) [33]. Stursova et al. (2012) also found that compared with Basidiomycota, Ascomycota are more involved in cellulose decomposition [34]. Meanwhile, Chytridiomycota and Glomeromycota, which had a lower relative abundance, also showed regular and significant changes over time (*p* < 0.05). Determining the role, if any, of these low-abundance microbes in responding to changes in plant diseases, soil health, and so on will be a fascinating challenge for future studies. The Venn diagram showed that 451 OTUs were shared by sugarcane during the growth periods (Figure 3B). The number of OTU unique to each of these periods varies considerably. To further explore, we used the source model of microorganisms to analyse the fungal transmission ratio between stages. Since more than 90% (91–95%, Figure 4B) of the fungal community were passed to the next period in each stage, it indicated that after the formation of the sugarcane rhizosphere fungal flora, although different microorganisms would be recruited or consumed at different periods, the overall structure would remain stable. Such a result also verifies the idea that plants can maintain resident soil fungal populations but not non-resident soil fungal populations, as has been previously verified in model plants [5].

### 3.2. Fungal Taxonomy, Function, and Soil Traits Were Interrelated, and the Transfer of Fungal Communities between Changes in Sugarcane Growth Periods Showed Regularity

Additionally, there was a strong correlation between soil environmental factors and sugarcane rhizosphere fungi populations in different growth stages. RDA and network map showed that TN, TK, pH, and AN were the main factors driving sugarcane fungal communities (Figure 4A,D). These soil nutrients directly or indirectly affected the survival and growth of sugarcane fungal communities, which is in agreement with the results reported by Zhang et al. (2016). They indicated that organic matter, total N and total P significantly affected soil fungal community composition in the southeastern Tengger Desert [35]. To further explore the influence of these nutrient factors on the fungal community during the growth period of sugarcane, we used the mantel test to calculate the correlation between three matrices and further validated that TN, TK, and pH were still the main environmental factors driving the OTU composition of the fungal community (Figure 4C). There were more genera that were decreasing in relative abundance in the rhizosphere over the life cycle of sugarcane, while fewer genera were increasing in relative abundance. It may imply the degree of sugarcane’s control over the rhizosphere fungal population under natural conditions. Additionally, the stabilization of the fungal population makes it difficult to re-enrich once it is lost from the rhizosphere environment, and the difficulty of plant recruitment to soil fungi also may cause this phenomenon (Figure 5D,E). This is similar to the results of tracking changes in the rice root microbial flora [28]. These data reinforce the separation role that the growth period plays in distinguishing plant rhizosphere microbial flora as observed in other studies [36,37,38], but it needs more research to illustrate. During the whole growth period of sugarcane, there are 10 fungal genera that have been decreasing, mainly including Ascomycota, Chytridiomycota, Eurotiales, Hypocreales, Sordariomycetes, and Tremellales. We speculate that the decline of these fungal genera is closely related to the genotype and developmental stage of sugarcane [39,40]. This requires us to conduct further verification. Such results provide a list of sugarcane rhizosphere fungi, which should now be targeted to elucidate their potential functions in the roots of sugarcane (symptomless colonizers vs. plant growth-promoters vs. pathogens) by targeted separation and sequencing technology. Furthermore, according to our results, *Pseudallescheria* had been enriched during the first three growth periods of sugarcane growth (Seeding–Elongation). However, *Nectriaceae* were significantly (*p* < 0.05) increased in the Tillering, Elongation and maturation stages (Figure 5A–C). Studies have reported that *Pseudallescheria* is a filamentous pathogenic fungus. Moreover, it is not only a potential human and animal pathogen, but also exists in the soil environment [41,42]. *Pseudallescheria* is of special importance for biological health and whether its significant variation in the rhizosphere could lead to local and disseminated infection of the host is a matter of alarm [43]. Studying the time trend of this pathogenic fungus in the sugarcane rhizosphere has great significance for the defense against certain new diseases that occur in the early growth of sugarcane in the future. Similarly, the ascomycete family *Nectriaceae* also includes numerous plant and human pathogens [44]. The enrichment of *Nectriaceae* is presumed to be potentially related to certain diseases that occurred in the late stage of sugarcane growth. However, evidence of pathogenicity does not necessarily exist, which needs to be established through more in-depth and mechanistic studies. Nonetheless, our findings revealed the most basic information and provided the possibility of sugarcane disease research. Additionally, it was found in the co-occurrence network that some fungal interactions between genera disappeared and then reappeared with time, presumably because the “autonomous consciousness” of sugarcane and environmental factors adjusted the balance of fungal interactions. The specific regulatory pathways and the types and sources of metabolites involved in them are not known. Whether this phenomenon is associated with the development of pokkah boeng disease remains to be proven.

### 3.3. Structural Equation Modeling Demonstrates a Correlation between Key Fungi, Soil Properties, and Sugarcane Disease Rates

Structural equation models (SEMs) indicated that soil temperature is a critical factor affecting the α-diversity of rhizosphere fungal communities of sugarcane, which showed significance in all models that we constructed (Figure 6 and Appendix A). Soil temperature changes triggered by seasonal changes had a certain perturbing effect on the rhizosphere community [45]. In addition, the strong direct negative effects of soil pH, AN and TN on the disease status of sugarcane suggest that soil infertile would increase the likelihood of sugarcane disease [46,47]. At the same time, *Fusarium* and *Talaromyces* also responded to changes in the soil nutrient environment, thus interacting adversely with the plant. The high-quality genome sequence of *Fusarium andiyazi* in China, published by Bao et al. (2021), provides more clarity on the role of *Fusarium* in sugarcane pokkah boeng disease [48].

### 3.4. Rhizosphere Fungal Community Assembly Is Mainly Influenced by Stochastic Factors

Furthermore, the results of the rhizosphere fungal community assembly process showed that the majority of the βNTI values among the samples were less than |2| (Figure 7B), indicating that the changes observed in the sugarcane rhizosphere fungal community over time were primarily influenced by stochastic factors [49], but the stochastic contribution increased and then decreased with sugarcane growth and development, eventually reaching the lowest value at the maturity stage (Appendix A). This is due to stochastic changes in the probability distribution and the relative abundance of species of rhizosphere fungi in the early stages of sugarcane growth (ecological drift) [50], and the formation of an adaptive system of root and soil environment in the later stages of sugarcane growth, resulting in abiotic and biotic factors playing an increasingly pivotal role in the presence or absence and relative abundance of rhizosphere fungi. Furthermore, mutual inhibition between rhizosphere fungi has an effect on community assembly, but the specific mechanism of action and whether it leads to a deterministic process of rhizosphere fungal communities needs to be supported by more research evidence. Phylogenetic trees and FUNGuild functional prediction analysis were applied to the study of sugarcane rhizosphere fungal communities to simplify and visualize the complex community functions and evolutionary relationships (Figure 8 and Appendix A). FUNGuild analysis revealed that the abundance of fungi showed differential responses across stages, for example, *Fusarium* was abundant in Seeding and Elongation and *Talaromyces* in Maturity. These results correlate with the work of Li et al. (2021), who reported that some potential biocontrol genera change with plant growth and tillage [51]. While showing dominant species richness, it also provided new ideas for seasonal prediction of rhizosphere pathogens of sugarcane and disease control during the different growth periods, such as annotated phytopathogens (*Cochliobolus*) that directly or indirectly contribute to the disease of sugarcane in a given stage.

## 4. Materials and Methods

### 4.1. Sugarcane Planting

In the spring of 2017, sugarcane cultivar, FN 41 was grown in a separate sugarcane experimental field in China to track the rhizosphere fungal population change procedure during the entire sugarcane growth cycle. Before planting, sugarcane seed-canes were soaked in 0.1% carbendazim (*w*/*v*) for 10 min to avoid surface-associated microbes followed by soaking in water for 24 h [52]. After soaking seed-canes, they were transferred to the fields at the Baisha experimental station (119°06′ E, 26°23′ N). Before the start of the experiment, the same variety of sugarcane crop was grown in the test plot for a period of three years. Abundant heat resources in the test area to supply sugarcane growth. The field growth of sugarcane was managed uniformly according to the farming conditions of local growers. A randomized group design was used, with an area of 1008 m^2^, 8 rows per zone, 5 m row length, 1.2 m row spacing, 48 m^2^ per plot, and 3 replications, distributed in randomized groups in each plot. The planting height of sugarcane was 1/2 of the row height. Twenty-one sugarcane stalks were randomly selected in each field and measured with a measuring tape and vernier caliper to determine stalk height and stem diameter in each period. Meanwhile, the number of effective plants, diseased plants and total plants were also counted.

### 4.2. Sample Collection

The experiment started on the 3 January 2017. The soil samples were collected at the Baisha Experimental Station of the National Sugarcane Research Center of Fujian Agriculture and Forestry University (subtropical monsoon climate, the annual average temperature is 19.5 °C, the annual average precipitation is 1673.9 mm). At different fertility stages, we used the “S-shaped sampling method” to select 21 sampling points in the sugarcane field [53]. According to the shaking-off method of Riley and Barber [54], the soil adhered to the sugarcane roots was brushed with a small sterile brush, and soil samples were finally collected, sieved at 2.0 mm, and stored in a refrigerator at −20 °C [55]. The samples were collected during the sugarcane seeding stage (Seeding 4, 2017), tillering period (Tillering 2, 2017), elongation period (Elongation 17, 2017), and maturity period (Maturity 29, 2017), with a total of 84 samples (4 periods, 21 repetitions).

### 4.3. Determination of Pokkah Boeng Disease of Sugarcane and Soil Physio-Chemical Properties

The physicochemical indicators in the rhizosphere soil that had a strong influence on the sugarcane and varied greatly during the succession of growth periods were selected for measurement. Soil suspension with water (1:2.5 WV^−1^) was prepared to estimate soil pH using a pH meter (PHS-3C, INESA Scientific Instrument Co., Ltd., Shanghai, China) [56]. The soil temperature (Tem) was measured with the Soil Temperature Detector (Model: JC-TW, Shandong, China). The available nitrogen was measured using the alkaline hydrolyzable diffusion method [57], Soil total nitrogen and organic matter were determined by Kjeldahl digestion and determined by the oil bath–K_2_CrO_7_ titration method [58,59]. C:N is the ratio of soil total N to organic matter. Soil total potassium and total phosphorus were determined by digestion with HF-HClO_4_, followed by flame photometry and molybdenum-blue colorimetry, respectively [60,61]. Available potassium was extracted by ammonium acetate and determined by flame photometry [62]. Available phosphorus was extracted by sodium bicarbonate and determined using the molybdenum blue method [63]. Field judgment of pokkah boeng disease of sugarcane was mainly based on the symptoms described in previous studies [64] and divided into three types: the chlorotic phase, the acute phase or top-rot phase and the knife-cut phase (associate with top rot phase).

The effective number of stems/ha = Effective number of stems/m^2^ × 666.67/average row spacing.

### 4.4. DNA Extraction and PCR Amplification

For 84 soil samples, DNA was extracted using a Power Soil DNA Isolation Kit (MoBio Laboratories Inc., Carlsbad, CA, USA) according to the manufacturer’s instructions. A NanoDrop 2000 spectrophotometer (Thermo Scientific, Waltham, MA, USA) was employed to assess the concentration and quality of DNA. Amplification of 18S rDNA gene fragments was carried out using primers set SSU0817F (5′-TTAGCATGGAATAATRRAATAGGA-3′)/SSU1196R(5′-TCTGGACCTGGTGAGTTTCC-3′) [65]. The reaction conditions used for DNA amplification were: 95 °C for 3 min, followed by 35 cycles of 95 °C for 30 s, 55 °C for 30 s, and 72 °C for 45 s, with a final extension at 72 °C for 10 min (GeneAmp 9700, ABI, Foster City, CA, USA). PCR reactions were carried out in triplicate in a 20 μL mixture containing 2 μL of 2.5 mM sNTPs, 4 μL of 5× Fast Pfu buffer, 0.4 μL of Fast Pfu polymerase, 0.4 μL of each primer (5 μM), and template DNA (10 ng). Extraction of amplicons was carried out using an AxyPrep DNA Gel Extraction Kit (Axygen Biosciences, Union City, CA, USA). Later, the DNA was quantified using QuantiFluor™-ST (Promega, Madison, WI, USA). Purified amplicons were pooled in equimolar and paired-end sequenced (2 × 250) on an Illumina MiSeq platform (Majorbio, Shanghai, China) following the standard procedures.

### 4.5. Data Quality Control and Filtering

The sequencing extracted data were saved in fastq format, while the paired reads were spliced (merged) into one sequence based on the overlap between PE reads, and quality control was performed on the read quality and merging effect. Valid sequences were obtained from the barcodes and primers at the first and last ends of the sequences, and then the double-ended sequences were spliced (Flash, 1.2.11) to generate an abundance table for each taxonomy (QIIME, 1.9.1). The complete data sets generated in our study have been deposited in the NCBI Sequence Read Archive database under BioProject ID PRJNA721464.

### 4.6. Sequences and Statistical Analysis

The UPARSE standard pipeline was used to analyze the sequence data [66]. Briefly, sequences with short reads (<250 bp) were filtered out before for downstream analysis. Sequences with ≥97% similarity were clustered into OTUs. All sequences were assigned using the RDP classifier to identify taxa with a confidence threshold of 0.8 [67]. We selected these OTU with 97% similarity, and then calculated the Alpha diversity index under different random sampling using Mothur [68]. Later, we used R to draw the rarefaction curves. The DPS software was used to analyze the variance of the soil physical and chemical properties, and the significance was calculated based on the Bonferroni test (FDR adjusted *p*  <  0.05) [69]. RDA was used to visualize the relationship between fungal communities and soil environmental factors. Network analysis was performed using R to calculate the correlation between the factors (Spearman correlation), and Cytoscape (version 3.6.1) was used to adjust and visualize the results [70]. Analysis of differential OTU abundance and taxa was performed using a DESeq2 of the R package, and then we used a Manhattan plot to visualize the results (R 3.6.0). Structural equation modeling (SEM) was performed using IBM SPSS Amos 26. Maximum likelihood estimation with standard errors was also used [71]. The nearest-taxon index (NTI) and βNTI (999 random) were used to quantify changes in rhizosphere fungal phylogeny over time, and the two indices were calculated using the package “picante” [72,73]. The heatmap and functional annotation and maximum likelihood trees were created using the majorbio platform (http://cloud.majorbio.com, accessed on 12 September 2022). FUNGuild classification map was completed using STAMP (version 2.1.1), comparing the two stages, using Welch’s t-test, 95% confidence intervals, *p* < 0.05. Network diagramming and parameter were calculated using R, Cytoscape and UCINET 6 together [74].

## 5. Conclusions

When focusing on soil productivity, rhizosphere microbiomes are a critical factor and closely related to plant health, especially the fungal community. Under the background conditions of this study, both the growth stages and soil characteristics of sugarcane had significant effects on the composition and function of the rhizosphere fungal community. The pH, TN, TK and AN decreased significantly during the growth of sugarcane. TN, TK, and AN were significantly decreased by 12.41%, 26.11%, and 31.32%, respectively, compared to the early growth stage. The dominant fungal phyla were Ascomycota, Basidiomycota, and Chytridiomycota, and the dominance of dominant fungal groups can be explained by both the temporal dynamics of sugarcane roots and the awareness of plant-autonomous regulation. We also discovered that diseases had significant and strong negative effects on selected soil traits (pH, Tem, and AN) using SEM, while soil temperature had a strong and direct positive effect on fungal α-diversity, and *Fusarium* and *Talaromyces* had some direct and indirect associations with sugarcane diseases. It indicates soil nutrients affected plant health and rhizosphere fungal interactions. The variation in rhizosphere fungal structure was mainly influenced by stochastic factors, and the decrease in stochastic contribution rate was hypothesized to be due to the coupling of sensitive fungal genera or modules in the rhizosphere fungal network with sugarcane growth dynamics to coordinate the growth process. These results contribute to the understanding of plant–rhizosphere fungal interactions and provide additional opportunities for the development of green agriculture and preventive tools for crop pests and diseases.

## Figures and Tables

**Figure 1 ijms-24-05701-f001:**
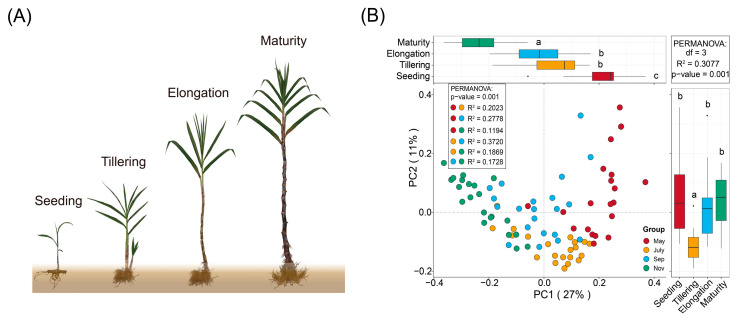
Fungal community composition under different sugarcane growth stages (**A**), as displayed by principal coordinates analysis based on Bray–Curtis distance (**B**). The growth stages of sugarcane are represented by Seeding, Tillering, Elongation, and Maturity stage, all *p*-values were 0.001, and different colors represented different stages.

**Figure 2 ijms-24-05701-f002:**
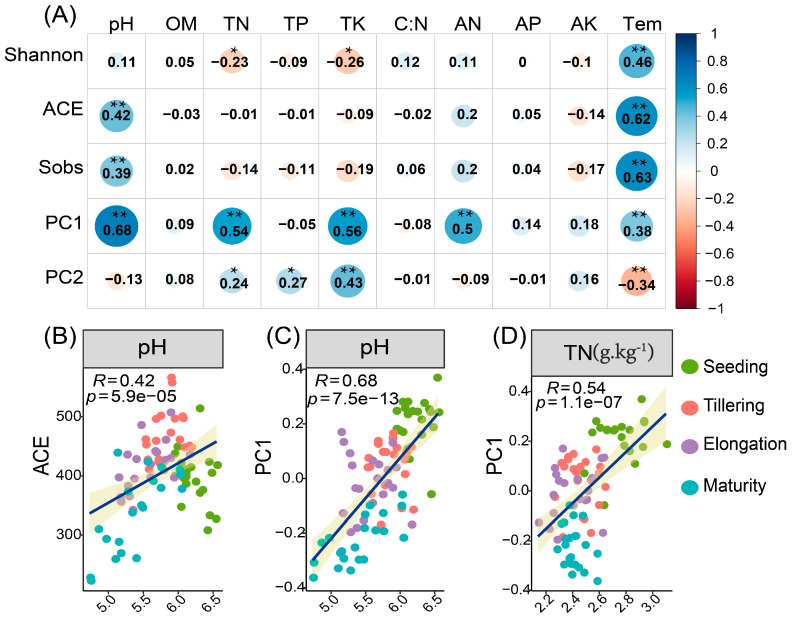
Pearson’s correlation coefficients for soil physio−chemical properties and fungal alpha diversity, PC1 and PC2 (**A**), and regressions analysis among soil environmental variables, fungal ACE (richness) and PC1 sampled during the stage of Seeding, Tillering, Elongation and Maturity, pH and ACE (**B**), pH and PC1 (**C**), TN and PC1 (**D**). * *p* < 0.05, ** *p* < 0.01.

**Figure 3 ijms-24-05701-f003:**
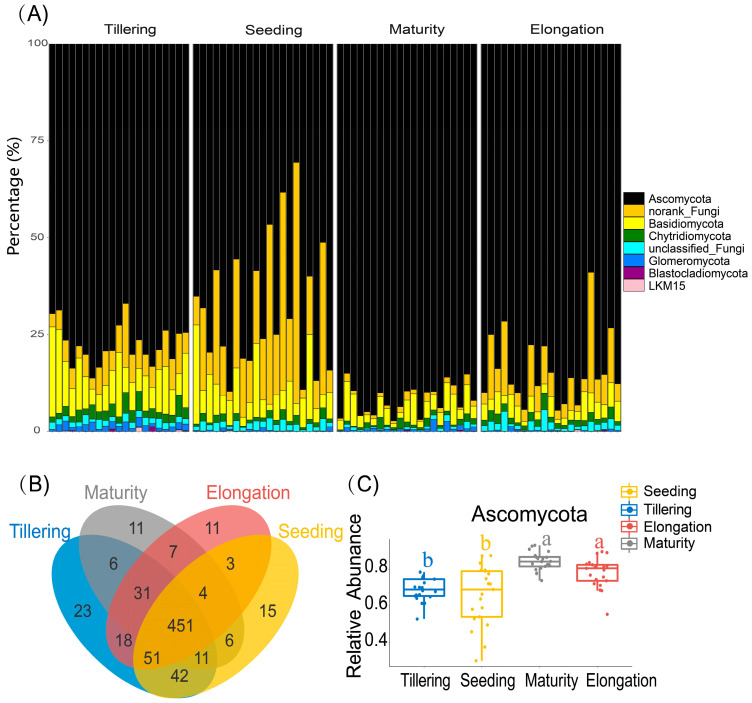
Different stages influence the diversity profiles of the rhizosphere soil microbiota. (**A**) The histogram showed that relative abundances of the main phyla according to four different growth periods were determined from twenty-one soil samples of each period. (**B**) A Venn diagram of operational taxonomic units (OTUs) among four stages (Seeding, Tillering, Elongation, Maturity). (**C**) The boxplot of the relative abundance of Ascomycota, the rhizosphere fungus of sugarcane, in different stages (FDR adjusted *p* < 0.05).

**Figure 4 ijms-24-05701-f004:**
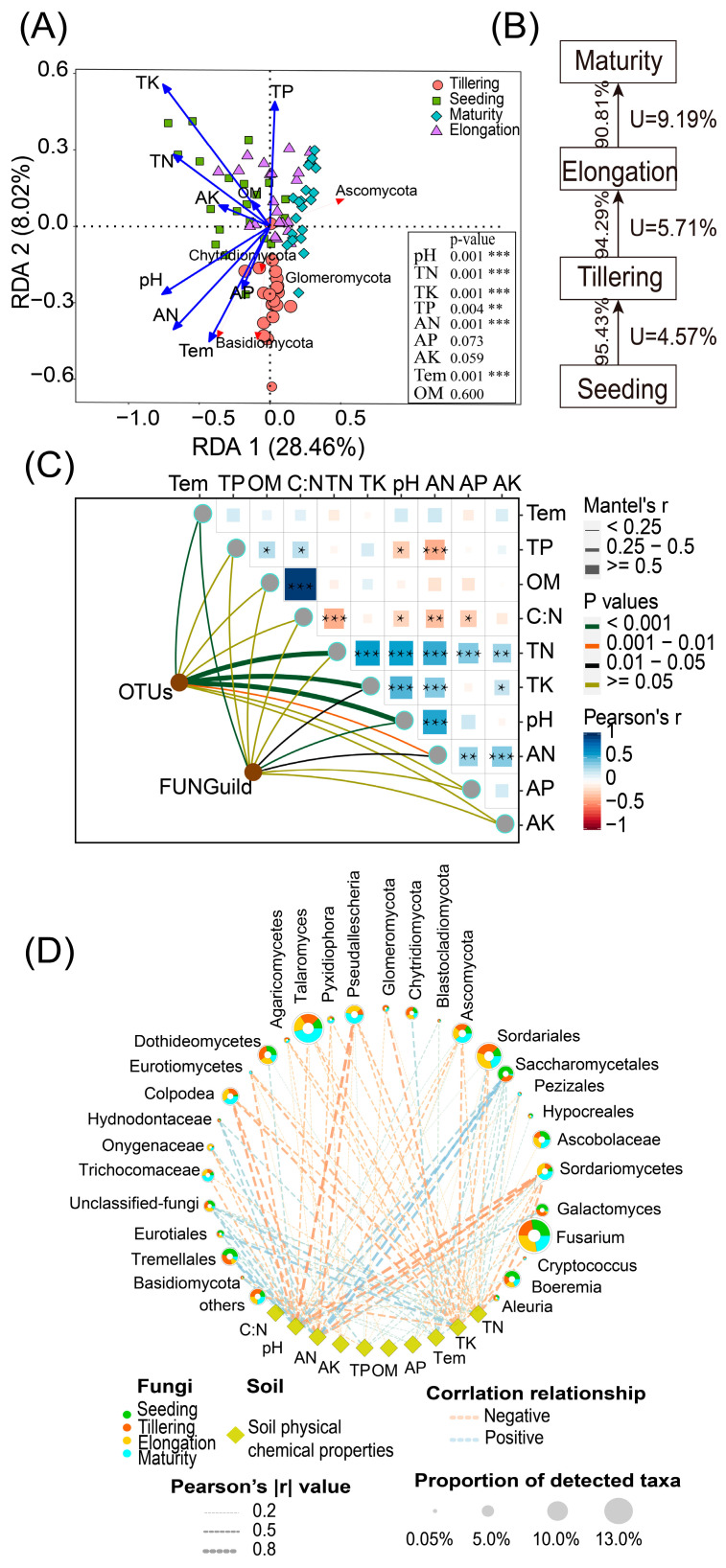
(**A**) Redundancy analysis (RDA) of soil properties and dominant Fungal phylum. In the part of the bottom right, the soil properties were fitted to the ordination plots using a 999 permutations test (*p*-values), significance: *** *p* < 0.001, ** *p* < 0.01, * *p* < 0.05. (**B**) Source analysis of sugarcane rhizosphere microorganisms. Source environment proportions for 84 samples of different stages were estimated using the Source Model of Plant Microbiome (SMPM). The number represents the proportion of fungal transmission in the adjacent period, and the letter U represents the proportion of microorganisms from unknown sources. (**C**). Environmental drivers of sugarcane rhizosphere fungal community composition. Pairwise comparisons of environmental factors are shown, with a color gradient denoting Pearson’s correlation coefficients. Edge width corresponds to the Mantel’s r statistic for the corresponding distance correlations, and edge color denotes the statistical significance based on 9,99 permutations. (**D**). Co-occurrence network of the mainly fungus genus population and soil properties in rhizosphere soils.

**Figure 5 ijms-24-05701-f005:**
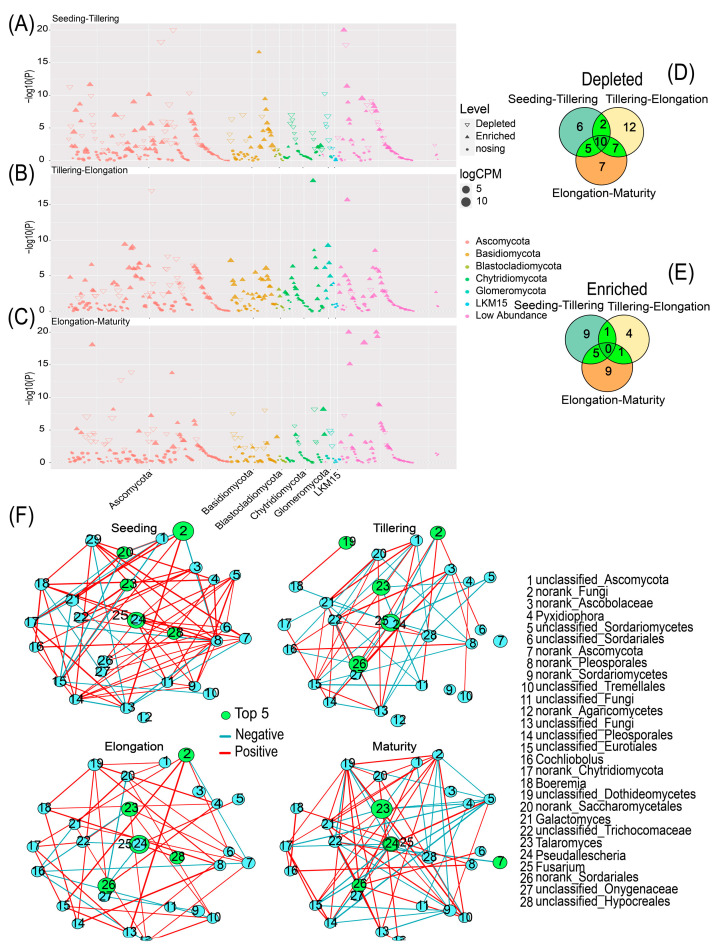
Taxonomic of differential fungi between the two periods of root microbiota. The Manhattan plot shows OTUs enriched or decreased in sugarcane in different stages. Each dot or triangle represents a single OTU. OTUs enriched or decreased in the sugarcane rhizosphere are represented by filled or empty triangles, respectively (FDR adjusted *p* < 0.05, Wilcoxon rank sum test). OTUs are arranged in taxonomic order and colored according to the phylum. (**A**) Seeding vs. Tillering; (**B**) Tillering vs. Elongation; (**C**) Elongation vs. Maturity. (**D**) Venn diagram of fungi decreasing between successive stages at the genus level. (**E**) Venn diagram of fungi enriched between successive stages at the genus level. (**F**) Co-occurrence network based on spearman correlation (Top 28 fungal genera, |r| > 0.4, *p* < 0.05), the size of the circles represents the relative abundance of the fungal genera and the thickness of the lines represents the size of the correlation.

**Figure 6 ijms-24-05701-f006:**
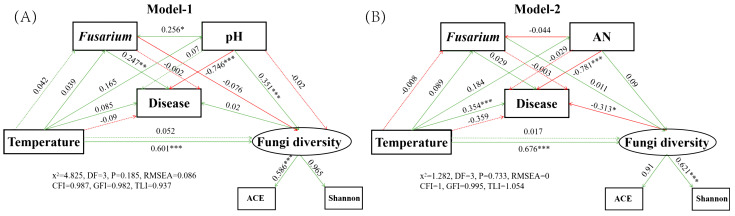
Structural equation modeling (SEM) analysis showed the contributions of Fusarium, temperature (soil temperature), soil and fungi diversity to the sugarcane disease. The disease was pokkah boeng disease of sugarcane. Soil was represented by mainly compositions including pH and AN (Available Nitrogen), and Model−1 (**A**) and Model−2 (**B**) were constructed respectively. Fungi diversity included ACE and Shannon index. The number above the arrow indicates the normalized path coefficient. *p*−value: * *p* < 0.05; ** *p* < 0.01; *** *p* < 0.001.

**Figure 7 ijms-24-05701-f007:**
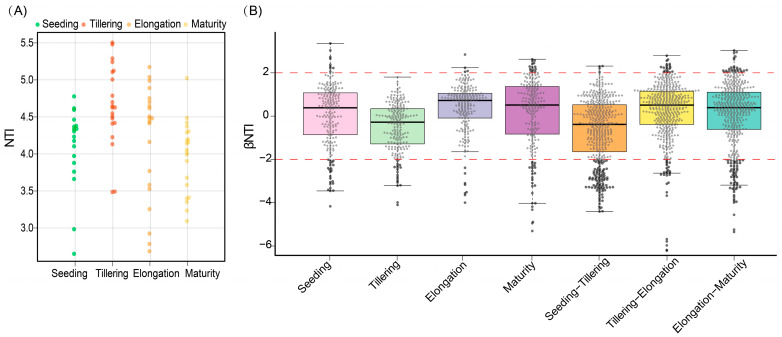
The distribution of NTI values (**A**) and box plots of β-NTI (**B**) for rhizosphere soil samples of sugarcane at different fertility stages, with different colors representing sample groups at different periods and red dashed lines marking the intervals of −2 and 2.

**Figure 8 ijms-24-05701-f008:**
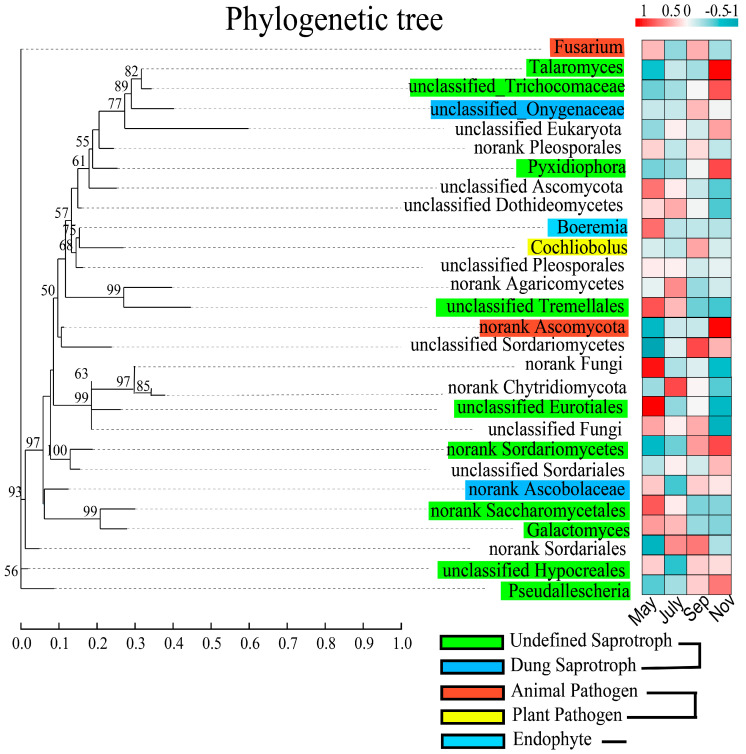
Phylogram with fungal guilds of the top 28 fungal genera, color the branches according to the phylum level to which the species belongs; maximum likelihood tree for the sequences obtained through high-throughput sequencing. The relative abundance data were normalized using the Z-score method, and then the average of the relative abundance of each genus for each group of samples was plotted as a heat map, the bar chart showed the percentage of Reads for the species in different stages. Guild annotation used the FUNGuild database.

**Table 1 ijms-24-05701-t001:** Changes in phenotypic indicators of sugarcane at different growth stages.

	Plant Height (cm)	Stem Diameter (mm)	Total Number/ha	Valid Number/ha	Effective Ratio (%)	Disease Ratio (%)
Seedling	42.49 ± 1.37 c	14.70 ± 0.80 c	3270	3185	97.40	1.4
Tillering	56.30 ± 2.10 c	23.70 ± 1.49 b	3473	3307	95.22	5.1
Elongation	122.90 ± 4.54 b	27.90 ± 1.25 b	2940	2103	71.52	19.8
Maturity	219.67 ± 5.98 a	36.67 ± 1.01 a	2950	2528	85.70	15.1

Note: Different letters in each column indicate significant differences among the stages at false discovery rate (FDR) adjusted *p* < 0.05. The data are presented in the form of mean ± standard error, the same below.

**Table 2 ijms-24-05701-t002:** Changes of soil nutrients in sugarcane at different growth stages.

	Seedling	Tillering	Elongation	Maturity
pH	6.22 ± 0.04 ^a^	5.83 ± 0.05 ^b^	5.55 ± 0.06 ^c^	5.38 ± 0.09 ^c^
OM/(g kg^−1^)	26.78 ± 0.89 ^ab^	25.03 ± 0.54 ^b^	30.80 ± 1.91 ^a^	26.71 ± 1.19 ^ab^
TN/(g kg^−1^)	2.74 ± 0.04 ^a^	2.45 ± 0.02 ^b^	2.39 ± 0.03 ^b^	2.40 ± 0.02 ^b^
TP/(g kg^−1^)	0.80 ± 0.01 ^b^	0.78 ± 0.01 ^b^	0.92 ± 0.02 ^a^	0.82 ± 0.01 ^b^
TK/(g kg^−1^)	23.25 ± 0.20 ^a^	16.58 ± 0.44 ^c^	19.65 ± 0.24 ^b^	17.18 ± 0.28 ^c^
AN/(mg kg^−1^)	182.06 ± 5.73 ^a^	168.75 ± 3.12 ^a^	127.01 ± 1.67 ^b^	125.04 ± 3.46 ^b^
AP/(mg kg^−1^)	78.61 ± 1.87 ^ab^	82.08 ± 1.19 ^a^	65.47 ± 1.48 ^c^	75.46 ± 2.39 ^b^
AK/(mg kg^−1^)	155.99 ± 4.08 ^a^	149.97 ± 3.49 ^ab^	136.11 ± 5.70 ^ab^	125.11 ± 14.72 ^b^
C:N	9.84 ± 0.35 ^b^	10.23 ± 0.27 ^b^	12.89 ± 0.80 ^a^	11.15 ± 0.52 ^ab^
Tem	24.49 ± 0.04 ^b^	30.36 ± 0.06 ^a^	30.22 ± 0.04 ^a^	19.37 ± 0.03 ^c^

Note: Organic Matter (OM), total nitrogen (TN), total phosphorus (TP), total potassium (TK), available nitrogen (AN), available phosphorus (AP) and available potassium (AK). C:N, the ratio of soil total nitrogen to organic matter. Tem: soil temperature. Different letters in each column indicate significant differences among the stages at FDR adjusted *p* < 0.05.

**Table 3 ijms-24-05701-t003:** Alpha diversity indices for different growth stages.

	Seedling	Tillering	Elongation	Maturity
Sobs	310.81 ± 6.34 ^b^	388.48 ± 6.79 ^a^	328.62 ± 6.46 ^b^	277.67 ± 12.15 ^c^
ACE	396.43 ± 9.88 ^b^	466.04 ± 9.49 ^a^	415.22 ± 9.08 ^b^	338.92 ± 14.47 ^c^
Shannon	3.36 ± 0.08 ^b^	3.83 ± 0.04 ^a^	3.52 ± 0.06 ^b^	3.35 ± 0.05 ^b^

Note: Different letters in each column indicate significant differences among the stages at FDR adjusted *p* < 0.05.

## Data Availability

The complete data sets generated in our study have been deposited in the NCBI Sequence Read Archive database under BioProject ID PRJNA721464.

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
