# Peer review of "Rhizosphere Fungal Dynamics in Sugarcane during Different Growth Stages"

_ijms, 2023, doi:10.3390/ijms24065701_

Round 1
Reviewer 1 Report
The manuscript entitled “Response of rhizosphere fungi to changes in growth periods of sugarcane'', has as the was to determine the drivers of the sugarcane fungal community, to understand the seasonal trends of these fungi and to explore fungi species closely related to changes in sugarcane growth period. The authors provide experimental evidence the results postulate more foundational insight into the process of sugarcane root microbiota exploration. The article was very well crafted and executed, just in need of minor revisions.
Minor suggestions:
The authors need to revise the title of the paper in a more meaningful way. The title is long and has some unnecessary information, suggestion: “Rhizosphere fungi dynamics in changes phenology stages of sugarcane”;
The abstract is written in a way lacks logic. It should highlight the salient findings more critically.
In table 1, 2 and 3, what do ± mean? the standard error with the standard deviation.
The results have long paragraphs. I suggest reducing the size of the paragraphs. The results of this study are not fully explained therefore the interpretation of the results is very difficult. The author needs to provide the % increase or decrease rather than just writing ''significantly increased….'';
The conclusion is totally confusing. Re-write the conclusion! It needs to be much improved.
Author Response
Reviewer 1
The manuscript entitled “Response of rhizosphere fungi to changes in growth periods of sugarcane'', has as the was to determine the drivers of the sugarcane fungal community, to understand the seasonal trends of these fungi and to explore fungi species closely related to changes in sugarcane growth period. The authors provide experimental evidence the results postulate more foundational insight into the process of sugarcane root microbiota exploration. The article was very well crafted and executed, just in need of minor revisions.
Minor suggestions:
- The authors need to revise the title of the paper in a more meaningful way. The title is long and has some unnecessary information, suggestion: “Rhizosphere fungi dynamics in changes phenology stages of sugarcane”;
Response: Thank you very much for your professional suggestion. The title you suggested is very helpful for us to improve the quality of the article. After consideration, we accept your suggestion and change the title of the article to "Rhizosphere fungi dynamics in changes phenology stages of sugarcane".
- The abstract is written in a way lacks logic. It should highlight the salient findings more critically.
Response: Thank you very much for your professional advice. We reorganized the specific content of the abstract, adding specific growth percentages of certain indicators and specific changes in rhizosphere fungi to highlight the important findings of this study. Specific revisions in the Abstract have been marked in yellow.
- In table 1, 2 and 3, what do ± mean? the standard error with the standard deviation.
Response: Thank you very much for your professional question, the information in the table is presented as mean ± standard error. We've added that information below the table. We apologize for the omission of this information. Thanks again for spotting our issue and pointing it out.
- The results have long paragraphs. I suggest reducing the size of the paragraphs. The results of this study are not fully explained therefore the interpretation of the results is very difficult. The author needs to provide the % increase or decrease rather than just writing ''significantly increased….'';
Response: Thank you very much for your professional advice. Based on your suggestions, we have changed the size of the original Results section by simplifying the language, providing percentage changes and restatements for different growth indicators. We have combined sections 2.2 and 2.3 from the original manuscript and provided a shorter description. Thanks again for your suggestion, it is of great significance for us to improve the quality of the manuscript.
- The conclusion is totally confusing. Re-write the conclusion! It needs to be much improved.
Response: We are very sorry for the confusing conclusions in the manuscript. We reorganized the language and changed the way of expression to concisely and clearly summarize the conclusions we obtained through this research. Thanks again for your very professional advice.
Reviewer 2 Report
ijms-2226913: ”Response of rhizosphere fungi to changes in growth periods of sugarcane”.
Research carried out by the authors seems to be important to the development and enhancement of existing information on this subject. Title is consistent with the content of work. Abstract, keywords are prepared in a clear manner and contain the necessary information. The methods used are appropriated. Tables and figures are informative. The work was constructed logically and the study contains a large amount of data. This paper is very interesting. But, some issues must be minor improved.
- There is no explanation of all the abbreviations used under the tables and figures.
- Fig.1 and Fig. 5 A-C are of difficult to read (it must be corrected).
- Please also, be sure that all the references cited in the manuscript are also included in the reference list and vice versa.
The paper can be accepted for publication after the aforementioned corrections have been made.
Author Response
Reviewer 2
ijms-2226913: ”Response of rhizosphere fungi to changes in growth periods of sugarcane”.
Research carried out by the authors seems to be important to the development and enhancement of existing information on this subject. Title is consistent with the content of work. Abstract, keywords are prepared in a clear manner and contain the necessary information. The methods used are appropriated. Tables and figures are informative. The work was constructed logically and the study contains a large amount of data. This paper is very interesting. But, some issues must be minor improved.
- - There is no explanation of all the abbreviations used under the tables and figures.
Response: Thank you very much for your professional advice. Regarding the abbreviations of physicochemical properties, we express the full names of the abbreviations in the description of the results. Based on your suggestion we have added the full names of all abbreviations below the table. Modifications made are marked in yellow.
- - Fig.1 and Fig. 5 A-C are of difficult to read (it must be corrected).
Response: Thank you very much for your professional advice. We have modified the Figure you mentioned, including changing the font and background color. In order to achieve the purpose of making it more convenient for readers to read.
- - Please also, be sure that all the references cited in the manuscript are also included in the reference list and vice versa.
Response: Thank you very much for your professional advice. Based on your suggestion, we rechecked and confirmed the citations of each reference and made revisions. Thank you for helping us refine and improve the manuscript.
The paper can be accepted for publication after the aforementioned corrections have been made.
Reviewer 3 Report
This manuscript presents a study on the dynamics of rhizosphere fungal communities associated with different growth stages of sugarcane using the Illumina sequencing platform. The results indicated maximum fungal diversity during the tillering stage and Ascomycota, Basidiomycota, and Chytridiomycota exhibited varied abundance during different stages of growth. Taxa belonging to Pseudallescheria and Nectriaceae were abundant during the three stages of sugarcane growth. Moreover, certain soil factors (pH, temperature, total nitrogen, and total potassium) significantly influenced the rhizosphere fungal community structure and structural equation modeling indicated a strong negative influence of soil factors on sugarcane disease status. From the results, it was concluded that the rhizosphere fungal communities during different stages of growth have a pivotal role in controlling potential fungal pathogens in sugarcane. Although the results are interesting, there are previous studies where changes in rhizosphere microbial community composition are reported in sugarcane. Further, there are certain concerns mentioned below that need attention.
1. The conclusions drawn in the study should be interpreted with caution as they are based on a single growing season using a sugarcane variety and a single soil type. This is a major shortfall in the study because previous studies have shown that observations for two or more growing seasons are necessary to elucidate any reliable trend in soil microbial community dynamics under field conditions. Further, earlier studies have also shown that the fungal community associated with a crop species can vary with crop varieties and soil types.
2. There is certainly critical information missing in the methods section. For instance, details on the cropping history of the experimental station, and cultural practices (fertilizer application, irrigation, weeding, etc.), etc., are missing. Further, climatic factors, the important determinants of plant growth under field conditions are also lacking in the present study.
3. Another important aspect that needs clarification is the actual process of soil sampling. Were the roots dugout or soil corers used to collect the soil samples during different stages of plant growth? Moreover, were sampling performed from the same or different plants during various developmental stages?
4. Is the sugarcane cultivar resistant or susceptible to pokkah boeng disease caused by Fusarium?
5. It is better to present the data and the results based on the growth stages of the sugarcane rather than the month of sampling.
6. Explain the abbreviation at their first mention and use them once they are introduced.
7. Italicize the scientific names in the reference list. Reduce the number of references, as they are too many for a research article.
8. Lines 2–3: Change the title to “Rhizosphere fungal dynamics in sugarcane during different growth stages”.
9. Lines 38–39: Avoid words that are already there in the title as keywords.
10. Line 64: Italicize the genus name.
11. Lines 77–79: Rhizosphere microorganisms not only solubilize potassium but also phosphorus.
12. Lines 82–83: The genus name in the binomial can be abbreviated after its first mention.
13. Line 90: Replace “a research object’ with ‘an experimental plant’.
14. Lines 95–96: The second and third objectives of the study sound similar.
15. Lines 105–107: This information should be in the methods.
16. Table 1: Is there any specific reason for presenting the values in the row for May in bold? Again why this row is separated from others by a line? Moreover,
17. Line 122: It is different growth stages and not fertility stages. Are the total numbers indicating the stem numbers? What are valid numbers and effective ratios?
18. Line 123: It should be ‘Different superscripts’. Similarly, tables should be self-explanatory. Therefore, avoid using abbreviations in the titles or explain to them if they are used.
19. Line 129: It should be ‘Different superscripts in a row…………….’ and not column.
20. Tables 1 and 2: It is better to name the plant growth stages (rooting/seedling, tillering, elongation, and maturity) than the sampling months.
21. Table 3: See comments for table 2.
22. Line 272: Italicize the genus name.
23. Lines 330–332: How could you a desert ecosystem with an agroecosystem when the conditions are entirely different?
24. Lines 358–360: What has the human pathogenic activity to do with the present study? Further, pathogenic fungi also tend to have non-pathogenic strains.
25. Lines 414–416: Rewrite the sentence for clarity.
26. Lines 417: Carbendiazim is a systemic fungicide and not a contact fungicide. Again, carbendiazim does not affect all microorganisms and is active only against fungi.
27. 424–425: What are effective plants? What was the method used to assess the disease incidence?
28. Lines 427–430: Provide the geographical coordinates of the experimental site.
29. Lines 433–434: I do not the need for sieving the soil samples with 2mm mesh when the soil attached to roots is not large. How were the 21 sampling points arrived at? Was any standard method https://doi.org/10.3389/fevo.2021.633155; https://doi.org/10.3791/57932) used to arrive at this number?
30. Lines 443–451: It is not necessary to reintroduce the abbreviation as they are introduced earlier.
31. Lines 459–460: Mention the quantity of soil used for DNA extraction.
32. Line 707: The term seedling refers to a plant originating from seed and not from cutting as in sugarcane.
33. Line 713: None of the probability values is in P<0.001 level.
34. Figure 1B: The PCA explains only less than 40% of the variation and more than 60% of the variations remain unexplained.
35. Figure 3C: What do the alphabets over the box plots indicate?
Author Response
Reviewer 3
This manuscript presents a study on the dynamics of rhizosphere fungal communities associated with different growth stages of sugarcane using the Illumina sequencing platform. The results indicated maximum fungal diversity during the tillering stage and Ascomycota, Basidiomycota, and Chytridiomycota exhibited varied abundance during different stages of growth. Taxa belonging to Pseudallescheria and Nectriaceae were abundant during the three stages of sugarcane growth. Moreover, certain soil factors (pH, temperature, total nitrogen, and total potassium) significantly influenced the rhizosphere fungal community structure and structural equation modeling indicated a strong negative influence of soil factors on sugarcane disease status. From the results, it was concluded that the rhizosphere fungal communities during different stages of growth have a pivotal role in controlling potential fungal pathogens in sugarcane. Although the results are interesting, there are previous studies where changes in rhizosphere microbial community composition are reported in sugarcane. Further, there are certain concerns mentioned below that need attention.
- The conclusions drawn in the study should be interpreted with caution as they are based on a single growing season using a sugarcane variety and a single soil type. This is a major shortfall in the study because previous studies have shown that observations for two or more growing seasons are necessary to elucidate any reliable trend in soil microbial community dynamics under field conditions. Further, earlier studies have also shown that the fungal community associated with a crop species can vary with crop varieties and soil types.
Response: Thank you very much for your very professional advice. We strongly agree with the issues you mentioned and we have revised the conclusion narrative based on your suggestions to emphasize the experimental background and limitations of this study and to correctly explain the scientific significance of this study.
- There is certainly critical information missing in the methods section. For instance, details on the cropping history of the experimental station, and cultural practices (fertilizer application, irrigation, weeding, etc.), etc., are missing. Further, climatic factors, the important determinants of plant growth under field conditions are also lacking in the present study.
Response: Thank you for your professional advice. Based on your suggestions, information on the climate of the sugarcane growing area and field management practices according to local sugarcane cultivation have been added to the Materials and Methods section.
- Another important aspect that needs clarification is the actual process of soil sampling. Were the roots dugout or soil corers used to collect the soil samples during different stages of plant growth? Moreover, were sampling performed from the same or different plants during various developmental stages?
Response: Thank you very much for your professional and detailed advice and questions. Plant inter-root soil samples were taken by digging out the sugarcane root system and destructively sampling. Different plants were sampled for different fertility periods. However, it was ensured that the plants taken were in the same growth period. Based on your suggestion in this section, we have added relevant information in the root sampling section.
- Is the sugarcane cultivar resistant or susceptible to pokkah boeng disease caused by Fusarium?
Response: Thank you often for your professional questions. In the context of the study of this experiment, it can only be determined that there is a correlation between the two, which is demonstrated in our structural equation model. However, to specifically determine the exact sensitivity of the two requires further experimental proof.
- It is better to present the data and the results based on the growth stages of the sugarcane rather than the month of sampling.
Response: Thank you very much for your professional advice. We have made the appropriate changes to the original results section containing month information based on your suggestions.
- Explain the abbreviation at their first mention and use them once they are introduced.
Response: Thank you very much for your professional advice. We have avoided the problem you raised by checking the abbreviations in the full text and explaining where they appear for the first time.
- Italicize the scientific names in the reference list. Reduce the number of references, as they are too many for a research article.
Response: Thank you very much for your professional advice. We have rechecked the content of the references and the issues related to the scientific names. In addition, we made sure that the references cited were required for the article.
- Lines 2–3: Change the title to “Rhizosphere fungal dynamics in sugarcane during different growth stages”.
Response: Thank you very much for your professional advice. We have revised the title to "Rhizosphere fungi dynamics in changing phenology stages of sugarcane" based on the title you provided.
- Lines 38–39: Avoid words that are already there in the title as keywords.
Response: Thank you very much for your professional advice, we have made appropriate changes in the keyword section to avoid the problem you mentioned.
- Line 64: Italicize the genus name.
Response: Thank you very much for your professional advice and we have corrected the error in the original article.
- Lines 77–79: Rhizosphere microorganisms not only solubilize potassium but also phosphorus.
Response: Thank you very much for your professional knowledge that has helped us to improve the quality of our manuscript, and we have revised the expression here in the original text to achieve a higher level of professionalism.
- Lines 82–83: The genus name in the binomial can be abbreviated after its first mention.
Response: Thank you very much for your detailed and professional advice, we have made appropriate changes here.
- Line 90: Replace “a research object’ with ‘an experimental plant’.
Response:Thank you very much for your professional advice. We have changed the sentence according to your suggestion and replaced the content you mentioned.
- Lines 95–96: The second and third objectives of the study sound similar.
Response: Thank you for the suggestion. We have re-narrated the study objectives in the original paper. The differences and innovations of the research objectives are highlighted. We are very grateful for the help your suggestions have brought to improve the quality of our manuscript.
- Lines 105–107: This information should be in the methods.
Response: Thank you very much for your professional advice, we have added this information in the methods section and revised the relevant description here.
- Table 1: Is there any specific reason for presenting the values in the row for May in bold? Again why this row is separated from others by a line? Moreover,
Response: Thank you very much for your meticulous advice, there is no special reason here. This one is a mistake in our table adjustment, thank you very much for pointing out our mistake, we have made the change in the original manuscript.
- Line 122: It is different growth stages and not fertility stages. Are the total numbers indicating the stem numbers? What are valid numbers and effective ratios?
Response: We apologize for our error in terminology and have corrected it to "growth stages". Thank you very much for your professional advice. total numbers refers to the total number of cane stems in the test plot, including effective stems and some canes that have fallen over or not grown up to the standard due to external factors. Effective stems refer to the number of cane plants above 1m in height per unit area at the maturity stage of sugarcane.
- Line 123: It should be ‘Different superscripts’. Similarly, tables should be self-explanatory. Therefore, avoid using abbreviations in the titles or explain to them if they are used.
Response: Thank you very much for your professional advice. For some of the abbreviations used in Table 2, we add the full name of each indicator at the bottom of the table. The abbreviations are included in the table to make the table more aesthetically pleasing. We apologize for the missing full name information and thank you for promptly raising the relevant errors.
- Line 129: It should be ‘Different superscripts in a row…………….’ and not column.
Response: Thank you very much for your detailed comments. The analysis of variance here is a comparison of the same indicator for different growth periods. Therefore we have used this tabular form.
- Tables 1 and 2: It is better to name the plant growth stages (rooting/seedling, tillering, elongation, and maturity) than the sampling months.
Response: Thank you very much for your detailed suggestions. We have changed the month information in the table to correspond the time to the different growing periods of sugarcane according to your suggestion.
- Table 3: See comments for table 2.
Response: We have also modified Table 3 accordingly based on your suggestions. Thank you very much for your professional and detailed advice.
- Line 272: Italicize the genus name.
Response: Thanks to your careful suggestions, we have fixed the error here and corrected it to italics.
- Lines 330–332: How could you a desert ecosystem with an agroecosystem when the conditions are entirely different?
Response: Thank you often for your professional questions, and here it is highlighted that in different ecosystems, nutrient content can greatly drive the composition of fungal communities. Therefore the soil properties that drive fungal communities in this study will be more convincing.
- Lines 358–360: What has the human pathogenic activity to do with the present study? Further, pathogenic fungi also tend to have non-pathogenic strains.
Response: Thank you very much for pointing out our error. This pathogenic fungus may be associated with unknown diseases arising in the future sugarcane production process. Therefore we have included this highly variable fungus as a potential subject. We have reworked the discussion in this section by removing statements that are not relevant to this study and rearranging the language. Thank you for your help in improving the quality of our manuscript.
- Lines 414–416: Rewrite the sentence for clarity.
Response: Thank you very much for your professional advice, we have reorganized the content and modified the original manuscript.
- Lines 417: Carbendiazim is a systemic fungicide and not a contact fungicide. Again, carbendiazim does not affect all microorganisms and is active only against fungi.
Response: Thank you very much for your professional advice. We will follow your advice in the subsequent trials to strictly operate and use more accurate and scientific fungicides.
- 424–425: What are effective plants? What was the method used to assess the disease incidence?
Response: Thank you very much for your question. Effective plants refer to the completion of the normal tillering process and the growth of tiller plants into independent sugarcane plants after a complete sugarcane growing period. The method of plant disease judgment is to count the number of diseased plants according to the characteristic description and picture comparison of plant diseases and pests in Guangxi.
- Lines 427–430: Provide the geographical coordinates of the experimental site.
Response: Thank you very much for your suggestion. Information about the geographical coordinates of the experimental stations has been provided in the section on sugarcane cultivation in our Materials and Methods. Your help in improving the quality of our manuscript is much appreciated.
- Lines 433–434: I do not the need for sieving the soil samples with 2mm mesh when the soil attached to roots is not large. How were the 21 sampling points arrived at? Was any standard method https://doi.org/10.3389/fevo.2021.633155; https://doi.org/10.3791/57932) used to arrive at this number?
Response: Thank you very much for your suggestion. The use of mesh screening can remove certain impurities while being able to obtain a more convenient soil for index determination, and also facilitate the subsequent DNA extraction, etc. Sampling points we are based on the S-shaped sampling method, 21 points are evenly distributed in the test plots to reduce the differences caused by different sampling locations, which can get more scientific results. Thank you very much for the references of the sampling method.
- Lines 443–451: It is not necessary to reintroduce the abbreviation as they are introduced earlier.
Response:Thank you very much for your suggestion, we have revised the section here in the original manuscript to remove some redundant abbreviations.
- Lines 459–460: Mention the quantity of soil used for DNA extraction.
Response: Thank you very much for your professional advice, we have added the content you mentioned in the DNA extraction section.
- Line 707: The term seedling refers to a plant originating from seed and not from cutting as in sugarcane.
Response: Thank you very much for your professional advice. For this experiment, sugarcane seedlings sprouted using seed stems with growth vigor near the tip of the sugarcane.
- Line 713: None of the probability values is in P<0.001 level.
Response: Thank you very much for your question. Our test of variance was subjected to a more rigorous test method (multiple comparisons, Bonferroni method), which led to a more rigorous screening of data variability and therefore did not appear. Under the conditions of LSD or Turkey test methods, there is a highly significant impact factor.
- Figure 1B: The PCA explains only less than 40% of the variation and more than 60% of the variations remain unexplained.
Response: Thank you very much for your question. This may be due to the high number of biological replicates we have, with 21 biological replicates per period. The variability between replicate individuals may cause the PCA analysis to extract principal components that do not completely summarize the overall variability, but on the time scale, samples from different growth periods are mostly completely separated.
- Figure 3C: What do the alphabets over the box plots indicate?
Response: Thank you very much for your question. Letters represent tests of variance according to the Bonferroni method (adjusted p-value less than 0.05), and different letters represent the presence of significant differences.
Round 2
Reviewer 3 Report
In this revised manuscript the authors have taken into consideration the suggestions and revised the manuscript accordingly. However, there are some issues that were not addressed.
1. Modify the title as shown in the annotated manuscript.
2. Present the results based on growth stages and not sampling months.
3. Do not reintroduce the abbreviations/symbols for the terms/elements that are introduced earlier.
4. Italicize all the scientific names in the reference list.

Author Response
- Modify the title as shown in the annotated manuscript.
Response: Thank you very much for your detailed and professional suggestions for revisions. These suggestions have led to a significant improvement in the quality of our manuscript. We have revised all the issues you have marked according to your suggestions. The title was changed to "Rhizosphere fungal dynamics in sugarcane during different growth stages".
- Present the results based on growth stages and not sampling months.
Response: Thank you very much for your suggestion. We have changed all month information in the article to growth stages. Such which includes changes to the content of Figure and changes in the supplementary material. In Figure, all information containing months in the original manuscript and supplementary material has been corrected.
- Do not reintroduce the abbreviations/symbols for the terms/elements that are introduced earlier.
Response: We have removed the abbreviations for the physical and chemical indicators of soil that were reintroduced in the materials and methods and removed the redundant information based on your suggestions. Thank you again for your detailed and professional suggestions for revision.
- Italicize all the scientific names in the reference list.
Response: We have italicized all scientific names in the references based on your suggestions. All changes are highlighted in blue.